behaviour/neuroscience/cognition

brain, cognition, brain size, bees, learning

**Author for correspondence:**
Miguel Á. Collado
e-mail: xmiguelangelcolladox@gmail.com

# Brain size predicts learning abilities in bees

Miguel Á. Collado[1,2], Cristina M. Montaner[1],
Francisco P. Molina[1,2], Daniel Sol[2] and
Ignasi Bartomeus[1]

[1]Estación Biológica de Doñana (EBD-CSIC), Avd. Americo Vespucio 26, 41092 Sevilla, Spain
[2]Centre de Recerca Ecològica i Aplicacions Forestals (CREAF-UAB), Campus de Bellaterra, Edifici C, 08193 Cerdanyola del Vallés, Spain

MÁC, 0000-0002-4216-317X; IB, 0000-0001-7893-4389

When it comes to the brain, bigger is generally considered better in terms of cognitive performance. While this notion is supported by studies of birds and primates showing that larger brains improve learning capacity, similar evidence is surprisingly lacking for invertebrates. Although the brain of invertebrates is smaller and simpler than that of vertebrates, recent work in insects has revealed enormous variation in size across species. Here, we ask whether bee species that have larger brains also have higher learning abilities. We conducted an experiment in which field-collected individuals had to associate an unconditioned stimulus (sucrose) with a conditioned stimulus (coloured strip). We found that most species can learn to associate a colour with a reward, yet some do so better than others. These differences in learning were related to brain size: species with larger brains—both absolute and relative to body size—exhibited enhanced performance to learn the reward-colour association. Our finding highlights the functional significance of brain size in insects, filling a major gap in our understanding of brain evolution and opening new opportunities for future research.

## 1. Introduction

A large brain is widely considered a distinctive feature of intelligence, a notion that mostly derives from studies in mammals and birds [1–4]. However, studies in insects demonstrate that cognitively sophisticated processes, such as solving complex problems by developing novel behaviours or learning new things from other individuals, are still possible with very small brains [5]. In fact, the million-fold increase in a large mammal's brain compared to an insect brain allows mammals to have behavioural repertoires not much bigger than insects [6].

If a large behavioural repertoire is possible with a miniature brain, what benefits are obtained by an animal investing in a

larger brain? Because brain size scales allometrically with body size [7], an explanation is that biophysical constraints force larger animals to have more and/or larger neurons [6]. It is for instance, easy to imagine that the bigger muscles of larger animals will require greater numbers of motor neurons and axons with larger diameters to cover longer distances [6]. More neurons may also allow greater replication of neuronal circuits, adding precision to sensory processes, detail to perception, more parallel processing and enlarged storage capacity [8]. These explanations are however insufficient because substantial variation in brain size remains even when the allometric effect of body size is taken out [9]. Given that neural tissue is extremely costly to maintain, what is the purpose of expanding the brain beyond allometric rules?

A plausible advantage of a disproportionately larger brain might be an enhanced ability to learn new behaviours to cope with novel or complex challenges [10,11]. While this hypothesis has received ample support from studies in birds and mammals [4,10,12], similar evidence is not available for small-brained animals like insects. Our insufficient understanding of the benefits of miniature brains remains thus a major obstacle for a general theory of brain evolution, and even casts doubts on whether variation in brain size is biologically meaningful [13].

In this study, we address this gap with an experimental comparative analysis in bees. Bees have historically fascinated biologists because of their small nervous systems compared to the complexity and diversity of their behaviour [6,14]. Numerous species have been reported to be able to create memories of rewarding experiences [15–17] as well as of punishment [18], and those memories can be retrieved at different times after learning, both in the short- and long-term [19]. The brain of bees has also experienced a remarkable evolutionary diversification, despite sharing the same brain architecture, and there is some evidence that such diversification is associated with ecological differences. For example, diet specialists tend to have bigger brains relative to body size than diet generalists, presumably because of the need to locate and remember target floral resources ([20], but see [21] and [22], for examples on other taxa).

We conducted a common garden experiment to measure learning performance in 120 wild individuals of 16 bee species captured in the wild, and then used a phylogenetic comparative framework to test whether species that performed better in the learning task had larger absolute and/ or relative brains. Characterizing learning abilities of wild insects with different life histories and ecologies for comparative purposes is challenging [6], but perhaps the most tractable way is to measure performance in an associative learning task. We used a novel quick-to-perform experimental method proposed by Muth *et al.* [23], which measures the performance of learning a reward-colour association. Associative learning is a cognitive process with high ecological relevance [19,24]. Although most associative learning work has been done in social bees, solitary bees also need to learn landscape landmarks and to recognize rewarding flower patches (e.g. [25]). Importantly, associative learning trials are short enough to be suitable for stress-intolerant species and facilitate standardization across species with varying life histories and ecologies, two major obstacles hindering past progress in linking brain size and learning improvement. The possibility of performing experiments directly from individuals captured in the field allows experimentation with non-model taxa, providing opportunities for broader comparative analyses of cognition.

# 2. Material and methods

## 2.1. Study subjects

We captured bees by hand netting ($n = 120$ individuals) from March to June 2018 in different open fields and urban parks from Andalusia, South of Spain. Bees were kept individually in vials within cold storage and transported to the laboratory, where they were transferred into separated transparent plastic enclosures for the trials (figure 1), which were conducted within 3 h of capture. The laboratory was at 20°C homogeneously illuminated. After the experiment, all individuals were identified at the species level by a taxonomist (F.P.M.), yielding a sample of 16 unique species from eight genera (electronic supplementary material, table S2).

## 2.2. Experimental apparatus

The experimental enclosures were 2.5 cm wide and 13 cm long transparent PVC prisms with ventilation holes and with removable perforated lids attached at both extremes (figure 1a). The enclosures were

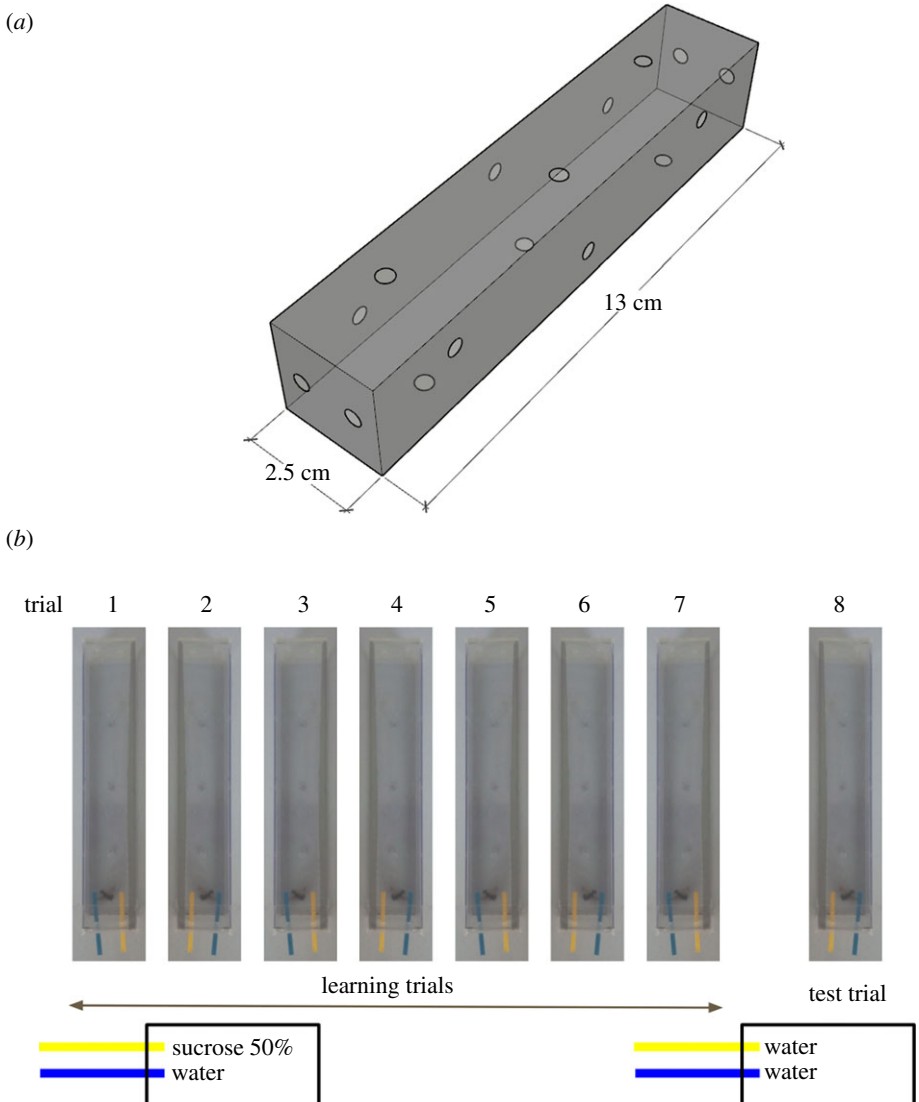

**Figure 1.** Experimental apparatus and design. (*a*) PVC experimental enclosures used for the experiment (2.5 cm wide and 13 cm long). Multiple holes were drilled for air circulation and easy strip offering from both extremes. (*b*) An example of the sequence of one complete set of trials for one single individual, where one colour is associated with a reward and it is maintained until the final test trial, where both strips are unrewarded.

placed on a grey surface during the experiment to minimize external stimuli. Enclosures were cleaned at the end of the experiments with water to avoid any scent cues afecting the trials.

## 2.3. Experimental trial

Associative learning was measured by a multi-choice free-moving proboscis extension protocol (FMPER, modified from [23]), where the animal had to learn to associate a reward (50% sucrose) with an arbitrary stimulus (a colour). Each experimental trial consisted of the presentation of yellow and blue cardboard small strips (3 × 0.2 cm) easily distinguishable by bees' vision [26,27]. The strips were always presented at the opposite extreme from where the individual was staying in the enclosure.

Before starting the trials, bees were left 30 min in the individual enclosures to allow them to recover from the cold and habituate to the experimental conditions. The experiment was divided into two phases, one where bees were trained to learn the task (seven *training trials*, hereafter), and another to assess whether they had learned the task (a *test trial*). During the training phase, one of the coloured strips was dipped in sucrose and the other one in water. The rewarded colour was randomly chosen but maintained during the whole experiment for each individual. The trial started when the individual reached the middle of the enclosure on its way towards the strips. We waited for the bee

to reach the strips and extend its proboscis to start drinking on one of them, waited for 3 s and removed the strip (figure 1b). We then allowed the bee to explore the remaining strip and again removed it after the individual had drunk for 3 s. Once this exploratory trial ended, the process was repeated every 10 min six more times to allow individuals to associate the stimuli (colour) with the reward (food) through associative learning (training trials), switching the strip's position to prevent bees learning to obtain the reward by using spatial information instead of colour cues (figure 1b). A trial was considered successful when the bee chose first the strip with sucrose and unsuccessful otherwise. The trial was considered finished when the subject drank from both strips or otherwise stopped after 2 min if no strip was touched.

After seven training trials, we tested 10 min later whether the individual had learned to associate a colour with a reward by means of an unrewarded test, where both strips were dipped into water. Learning was defined in terms of success or failure in solving the test. Across all trials, when solved, we also quantified the time needed to start drinking from the correct strip (latency, hereafter). While absolute values of latency to solve the task may be more influenced by motivational states or body size, the rate of change in this latency can still give valuable information complementary to the learning success measure.

Thus, the only modification of our experimental design from the original FMPER protocol [23] is that we removed the first acclimatization trials and considered the first trial as acclimatization/exploration. In general, bees responded well to the experimental procedure, especially those from genus *Bombus*, *Lasioglossum* and *Andrena*, and the species *Apis mellifera*, and *Rhodanthidium sticticum*. However, in addition to the 120 bees analysed here, another 45 individuals from the genus *Anthophora, Eucera* and *Xylocopa* either ignored or did not react to a full experimental procedure (electronic supplementary material, table S2), therefore, these species were not included in the analyses.

## 2.4. Brain measurements

After the experiment, bees were anesthetized in cold chambers [28] and decapitated. The head was fixed in 4% paraformaldehyde with phosphate buffer saline. The fixative solution embedded the brain and dehydrated the tissue, preventing the brain from degradation for a long period. Although fixation can produce some brain shrinkage [29], we processed all brains in the same way and hence we assume that shrinkage was equivalent across species. Brains were extracted from the head capsule, separated from the tracheas and fat bodies to avoid weighting errors, and placed on a small piece of tared Parafilm®. Fixative solution was dried from the brain using Kimwipes® tissues and then the brain was weighted in a microbalance to microgram accuracy (Sartorius Cubis®). Brain weight was used as a proxy of brain size, as it is strongly correlated with brain volume of the mushroom bodies (correlation coefficient = 0.85; p-value < 0.001, [20]), which are the neuropil centres of most cognitive abilities [30]. Body size was measured as the intertegular distance, that is, the distance between the wing bases, usually used as a proxy of body size in bees [31].

## 2.5. Data analysis

A trial was considered successful when the bee chose first the strip with sucrose (proboscis extension) and unsuccessful otherwise. For each trial, we also measured the time needed for the individual to reach the correct strip and extend its proboscis to start drinking. These two metrics—success and latency in choosing the correct strip—were modelled by means of Bayesian generalized linear mixed models (BGLMM), as implemented in the package *brms* [32]. Because models include several individuals of each species, we treated species as a random factor in all models. In addition, we incorporated a phylogenetic covariance matrix to control for non-independence of data owing to common ancestry. The phylogeny used was a maximum-likelihood phylogenetic tree of the superfamily Apoidea at the genera level ([33]; figure 2). Owing to the absence of infrageneric phylogenies for our genera, we simulated infrageneric polytomies within our phylogeny. Species tips were added to the phylogenetic tree genera nodes as polytomies of equal branch length relative to the genera branch length [31] using the phytools package (v. 0.6-44; [34]).

We first assessed whether individuals learned to associate a colour with a reward. To that end, we calculated the probability of success in the test trial by running a BGLMM without predictors, but with phylogeny and species as random effects. To further investigate the learning process, we asked whether success in choosing the correct strip increased over time (i.e. over the trials) and whether the latency to choose the correct strip declined over time. Note that by controlling for species in the

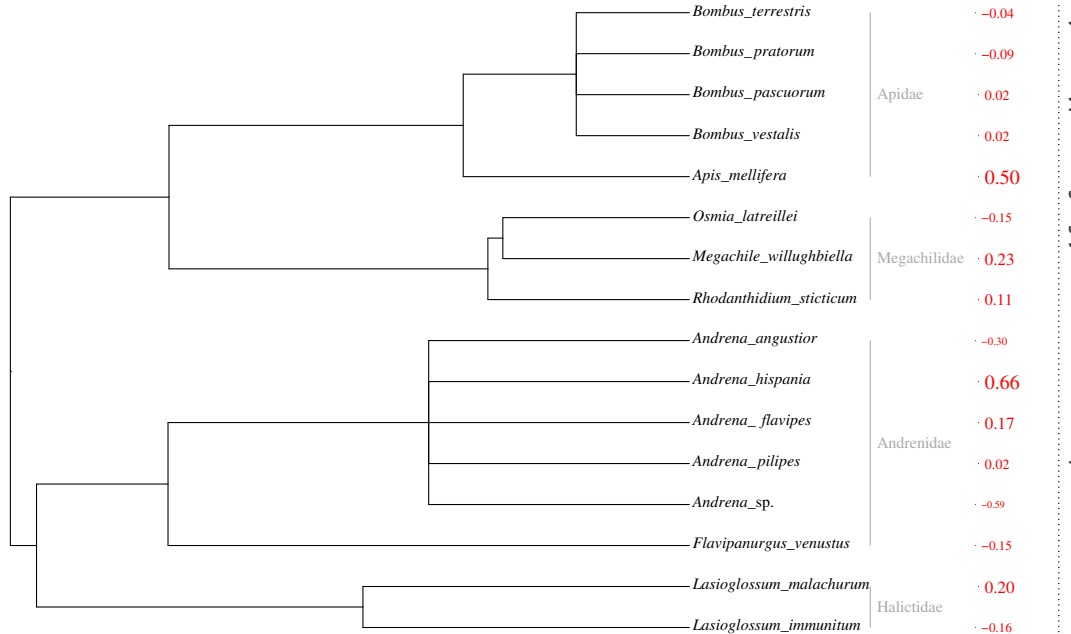

phylogenetic tree:
body size – brain weight residuals

**Figure 2.** Phylogenetic tree of the studied species. Numbers in red represent body size—brain size residuals for each species. Values above zero have bigger brains than expected by body size, and values below zero have smaller brains than expected.

random structure of our models, we do not confound differences in absolute latency across species (i.e. the random intercept), which may be driven by other factors such as body size, with the rate of change of the latency to succeed (i.e. the estimated slope). We modelled successes and failures (binary response) as a function of time (trial order) by means of BGLMMs with a Bernoulli structure of errors. As the same individual was tested along different trials, we included a code for each individual as a random factor in addition to species and phylogeny. Latencies were analysed in a similar way, but implementing BGLMMs with negative binomial error structures. Next, we examined whether individuals that better learned to associate colour and reward during the learning phase exhibited a higher probability of successfully solving the unrewarded stimulus test. This was modelled by testing if success in the learning test (binary response) was related to the number of correct choices during the seven training trials. Successes and failures were again modelled by means of BGLMMs with a Bernoulli structure of errors, including species and phylogeny as random effects.

Prior to the following analysis, we assessed the assumption that brain size is conserved phylogenetically by calculating the intraclass correlation coefficients (ICC; [35]) of a null model of brain size (both in absolute terms and controlling for body size, respectively) with no predictors, but including species and phylogeny as explained above.

We next analysed whether bees with larger brains were better at learning. We did so in two ways. First, we used a BGLMM with a Bernoulli error structure to ask whether the probability of successfully solving the unrewarded stimulus test was higher for species with larger brains, again accounting for variation within species and across the phylogeny. Further, we also used a BGLMM with a Gaussian structure of errors to evaluate whether the total number of successful trials during the learning phase was higher for species with larger brains, again accounting for variation within species and across the phylogeny. Following previous studies [36,37], we analysed brain size in both absolute terms (brain weight) and relative to body size. Relative brain size was estimated as brain size residuals [38,39], extracted from a Gaussian model with log-body size as fixed effect and species and phylogeny as random effects. To evaluate their effect on total number of successful trials, absolute and relative brain size were fitted as fixed continuous predictors in separated models including individual, species and phylogeny as random effects.

Our second approach to assess whether bees with larger brains are better at learning was to ask whether the latency to choose the correct strip decreased faster along the trials for species with larger brains. To that end, we used a BGLMM with a negative binomial structure of errors where we modelled the latency to touch the correct stripe as a function of trial (i.e. time), brain size (absolute or

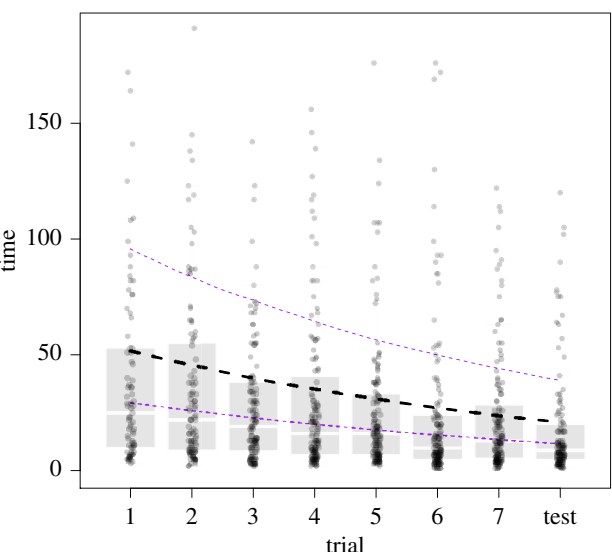

**Figure 3.** Changes in the latency (seconds) until touching the rewarded strip over the training trials. Dots represent each individual success. We superimposed to these data points the distribution per trial by means of a boxplot that depicts the median and 2nd and 3rd quantiles of the data. We plot the estimate and confidence intervals of the BGLMM negative binomial model ($\beta = -0.14 \pm 0.02$, confidence interval $= -0.18$–$0.10$). Last trial was considered the test as it was not rewarded.

relative, in separated models), and their interaction. The interaction term will indicate if the speed at which latency decreases with time is different for different brain sizes. Again, we account in the model for variation within individuals, species and across the phylogeny.

## 3. Results

Most bees learned to associate a colour with a food reward. First, the mean probability of an individual choosing the correct strip in the test trial was $0.64 \pm 0.15$ (binomial BGLMM, accounting for species and phylogeny), which is above the 0.5 probability expected by chance. Second, bees had more chance of success in the later trials than in the earlier ones (BGLMM, Bernoulli: $\beta = 0.17 \pm 0.03$, ICC $= 0.11$ to $0.24$) and decreased the latency to touch the correct strip along the trials (BGLMM, negative binomial: $\beta = -0.14 \pm 0.02$, ICC $= -0.18$ to $-0.10$; figure 3 and table 1). This suggests that individuals tended to improve in the task over time. Removing the first trial, which can be merely exploratory, did not change the above conclusions. Finally, individuals that chose the correct strip more often during the trials were more likely to succeed in the learning test (BGLMM, Bernoulli: $\beta = 0.34 \pm 0.14$, ICC $= 0.07$ to $0.62$).

However, there was substantial variation across species in the probability of choosing the correct strip, implying that not all species had learned the task. We asked whether learning differences across species reflected in part variation in brain size. We found evidence that bees with bigger brains—both when measured in absolute terms or relative to body size—were more likely to learn. On one hand, they had higher chances of success in the test trial (for absolute brain size BGLMM, Bernoulli: $\beta = 3.18 \pm 1.31$, ICC $= 0.82$ to $6.08$; for relative brain size BGLMM, Bernoulli: $\beta = 4.35 \pm 2.64$, ICC $= -0.80$–$9.87$; figure 4a,b) and a higher number of successful trials (including training and test, for absolute brain size BGLMM, Gaussian: $\beta = 0.49 \pm 0.22$, ICC $= 0.04$–$0.91$; for relative brain size BGLMM, Gaussian: $\beta = 3.53 \pm 1.92$, ICC $= -0.26$–$7.38$; figure 4c,d). On the other hand, the latency to learn decreased faster over time for bees with larger absolute brains (BGLMM, negbinomial, interaction absolute brain size: trial: $\beta = -0.05 \pm 0.01$, ICC $= -0.08$ to $-0.02$; figure 5a and table 2), but not for bees with larger relative brain sizes (BGLMM, negbinomial, interaction relative brain size: trial: $\beta = 0.35 \pm 0.18$, ICC $= -0.01$–$0.71$; figure 5b). These results held even when accounting for the fact that brain size was highly conserved phylogenetically (phylogenetic heritability [Ph]; $Ph_{Absolute\ brain} = 0.77$; $Ph_{Relative\ brain} = 0.83$).

## 4. Discussion

Highly controlled laboratory experiments allow the testing of detailed learning abilities, but only for a handful of species (e.g. honeybees, bumblebees, cockroaches, fruit flies) that can be raised in

**Table 1.** Results of the Bayesian models of learning, and learning related to brain size. (CI, confidence interval; $\beta$, estimate ± s.e. All models include species as random factors.)

| formula | $\beta$ | CI | notes |
|---|---|---|---|
| **learning** | | | |
| BGLMM Bernoulli: trial success ∼ trial number | 0.17 ± 0.03 | 0.11 to 0.24 | all trials (including test) considered |
| BGLMM negative binomial: latency ∼ trial number | −0.14 ± 0.02 | −0.18 to −0.10 | all trials (including test) considered |
| BGLMM negative binomial: success in learning test ∼ number of successful training trials | 0.34 ± 0.14 | 0.07 to 0.62 | first trial not considered |
| **learning relative to brain sizes** | | | |
| BGLMM Bernoulli: learning success ∼ absolute brain size | 3.18 ± 1.33 | 0.82 to 6.19 | just learning test |
| BGLMM Bernoulli: learning success ∼ relative brain size | 4.35 ± 2.64 | −0.80 to 9.87 | just learning test |
| BGLMM Gaussian: number of successful trials ∼ absolute brain size | 0.49 ± 0.22 | 0.04 to 0.91 | all trials (including test) considered |
| BGLMM Gaussian: number of successful trials ∼ relative brain size | 3.53 ± 1.92 | −0.26 to 7.38 | all trials (including test) considered |

laboratory conditions. Some of these experiments often involve stressful conditions, like individuals being fully harnessed in proboscis extension reflex (PER) protocols (see [40]). Therefore, laboratory experiments are only suitable for highly stress-tolerant species. The multi-choice FMPER [23] extends this experimental paradigm to conduct quick-to-perform tests of learning abilities in species directly captured from the field, some of them never tested before. This novel quick-to-perform experimental method allowed us to conduct a comparative analysis including multiple bee species of a wide array of ecologies captured directly from the field. Using this comparative framework, we found that most bee species, including solitary species never used before in cognitive experiments, can learn to associate a colour with a reward. Interestingly, species differed in their learning abilities, and these differences were in part explained by brain size. We show that the probability to learn increased with both absolute and relative brain sizes, albeit learning improvement along the trials show a more complex pattern and is related positively with absolute brain size, but not to relative brain size.

Our results suggest that the probability to learn increased with both absolute and relative brain sizes, albeit learning improvement along the trials show a more complex pattern and is only positively related with absolute brain size. Relative brain size and absolute brain size are two metrics with different assumptions and interpretations [7,13]. The finding that absolute brain size is related to learning performance may suggest that just having a larger amount of neural tissue can improve learning abilities [37], at least within our studied taxa. However, larger bees also have larger visual organs and higher mobility [31,41], which can facilitate the learning task for non-cognitive reasons. More interesting is the finding that relative brain size is related to learning ability. Thus, the probability of performing the learning task was higher in large-brained bees than in small-brained bees and, although both groups learned the task at similar speed, bees with smaller relative brains made more errors in the first trials. These results suggest that investing in extra neural-tissue beyond the allometric requirements pays off in terms of learning abilities ([6]; but see [13]).

While the use of wild animals allows broader interspecific comparative analyses, the approach may also have some caveats. Stress may make individuals behave in strange ways [42]. In fact, a handful of species did not react to the experimental settings, showing no interest for the coloured stripes. Specifically, we found 11 species that did not react to any complete experimental protocol (i.e. were not active in enough trials to consider a valid test) and six more species that fully ignored the experimental setting. However, for most of this species, very few individuals were tested. The lack of responsiveness during the experiment may also reflect neophobia [43], lack of motivation [44] or

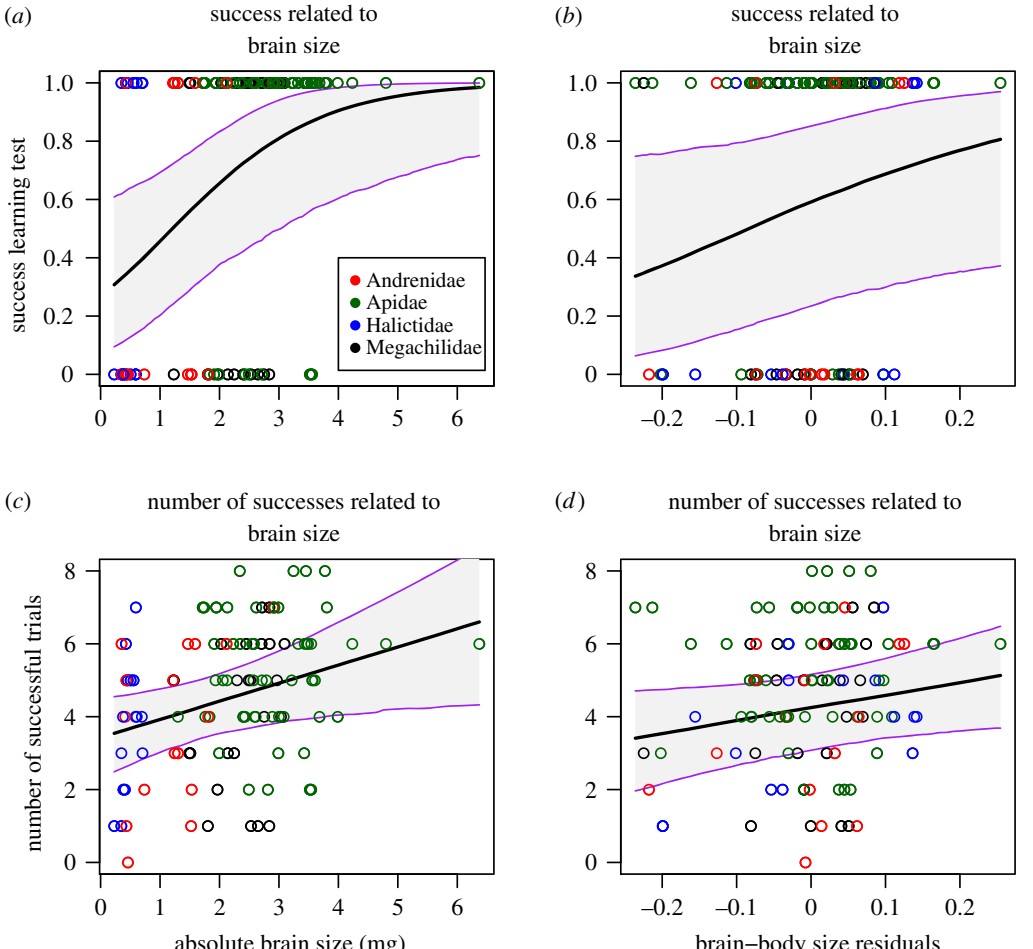

**Figure 4.** Relationship between brain size and learning performance. (*a*) Probability of success in the test trial as a function of absolute brain size (brain weight in mg); (*b*) probability of success in the test trial as a function of relative brain size; (*c*) relationship between the number of successful trials along the experiment and absolute brain size; and (*d*) relationship between the number of successful trials along the experiment and relative brain size. Overlapping lines represent the trends extracted from the BGLMM models and their CI. Individual data points are colour-coded according to the bee family they belong to visualize the phylogenetical relationships. The number of individuals for each family were: Andrenidae 18, Apidae 55, Halictidae 23 and Megachilidae 24.

although unlikely given the colour choice, it can also reflect a poor colour perception by some bee species [45]. Therefore, species that did not react were not necessarily 'species unable to learn' and alternative explanations are possible. Using wild animals can also have caveats, as they can vary in respect to their previous experience. However, evidence from *A. mellifera* does not suggest that using wild individuals change the results of learning tests compared to bees reared in the laboratory [23], but further analyses are needed for species that do not habituate well to experimental settings. Although this experimental approach was not suitable for testing all species, the number of candidate species we can test increases dramatically in comparison to classic experimental methods such as the PER which is mostly conducted on *A. mellifera* and bumblebees [18,24,40]. Expanding the range of species evaluated by applying quick-to-perform experimental protocols to wild species can provide important insights into the relationship between brain size and learning. Although our study was restricted to a particular learning domain, and testing correlations with general brain size measures rather than brain architecture, the relationships found are promising and call for a more detailed understanding of how brain size and architecture affect learning perfomance in different insect species. For example, although our experiment was not designed to analyse differences across social and solitary species, we observed that some solitary species can show similar learning abilities to social species (e.g. *Bombus terrestris*, *A. mellifera*, electronic supplementary material, figure S2). This appears to contradict the traditional view that social bees have more complex learning abilities than solitary bees [46], although more research is needed to confirm it.

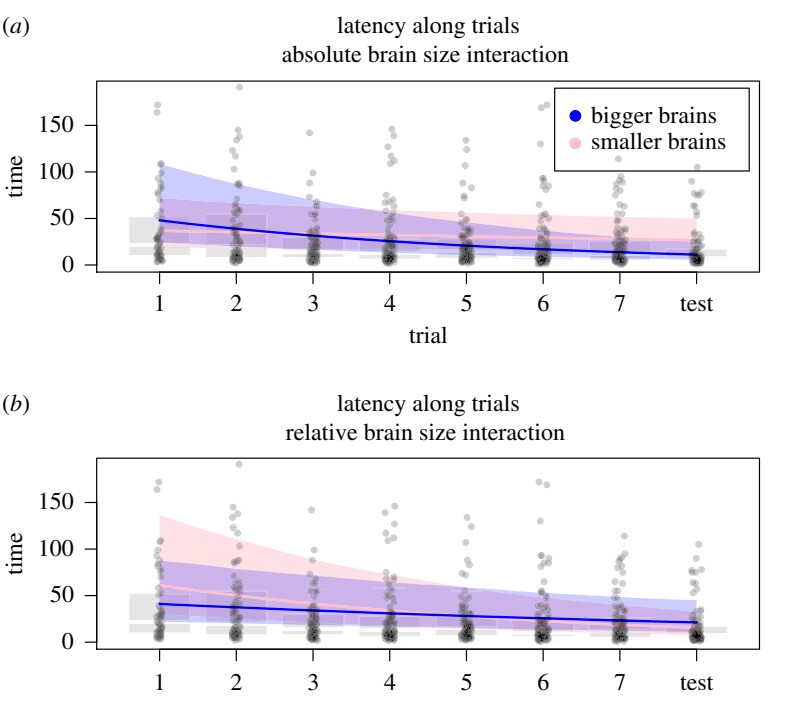

**Figure 5.** Latency (seconds) to reach the rewarded strip (or the correct but unrewarded strip in the test) predicted for different brain sizes categories. All analyses are carried out with brain size as a continuous variable, we cannot plot an interaction term between two continuous variables. In order to visualize this interaction, we show the predicted slopes when fixing one of the variables (i.e. brain size) to two representative values corresponding to the X and Y percentiles for both absolute brain size and relative brain size.

**Table 2.** Results of the Bayesian models of latency related to brain size, both absolute brain size and relative brain size. (β, estimate ± s.e; CI, confidence interval. All models include species as random factors, phylogeny and individual.)

| latency relative to brain sizes | | | | |
|---|---|---|---|---|
| formula | parameter | β | CI | notes |
| BGLMM negative binomial: | trial | −0.03 ± 0.04 | −0.10 to 0.05 | all trials |
| latency ∼ trial: absolute brain size | absolute brain size | 0.09 ± 0.12 | −0.13 to 0.33 | all trials |
| | trial: brain size | −0.05 ± 0.01 | −0.08 to −0.02 | all trials |
| BGLMM negative binomial: | trial | −0.14 ± 0.02 | −0.17 to −0.11 | all trials |
| latency ∼ trial: relative brain size | relative brain size | −1.70 ± 1.08 | −3.83 to 0.38 | all trials |
| | trial: brain size | 0.35 ± 0.18 | −0.01 to 0.71 | all trials |

Our results provide, to our knowledge, the first evidence that in insects, brain size matters in terms of learning performance, challenging previous claims that variation in brain size is not biologically meaningful ([6,13], but see [47]). However, it remains to be demonstrated whether similar patterns can be extended to other learning mechanisms. The underlying processes also warrant explanation. The challenge is to elucidate whether variation in learning performance across species reflects sensorial, cognitive, physical or emotional responses, and how these responses are associated with finer brain structures like mushroom bodies, neuron density or optimized neuron synapses.

Data accessibility. Data and relevant code for this research work are stored in the Dryad Digital Repository https://doi.org/10.5061/dryad.mpg4f4qxw [48] and have been archived within the Zenodo repository https://doi.org/10.5281/zenodo.4633103.

Authors' contributions. M.A.C. and I.B. worked on the original idea and designed the experiment. M.A.C., C.M.M. and F.P.M. captured the bees, all identified by F.P.M. M.A.C. and C.M.M. did the experiments and gathered the data.

M.A.C., D.S. and I.B. did the analysis and M.A.C. wrote the first draft; and I.B. and D.S. contributed to improve the manuscript.

Funding. This work was supported by MINISTERIO DE ECONOMÍA Y COMPETITIVIDAD, GOBIERNO DE ESPAÑA for the projects grant/award nos: CGL2013-47448-P and CGL2017-90033-P.

Competing interests. We declare we have no competing interests.

Acknlowledgements. We want to thank Felicity Muth & Anne S. Leonard for their inspiring work on FMPER and Carlos Zaragoza, for his help doing some of the experiments. We are grateful to Consejería de Medio Ambiente, Junta de Andalucía, for permission to work in Sierra de Cazorla and providing invaluable facilities there, and to EBD-CSIC for making Roblehondo field station available to us.

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
