## [Peer Review File · Royal Society Open Science]

Review History

RSOS-201940.R0 (Original submission)

Review form: Reviewer 1

Is the manuscript scientifically sound in its present form?

Yes

Are the interpretations and conclusions justified by the results?

Yes

Is the language acceptable?

Yes

Do you have any ethical concerns with this paper?

No

Have you any concerns about statistical analyses in this paper?

No

Recommendation?

Accept with minor revision (please list in comments)

Comments to the Author(s)

This manuscript addresses the longstanding hypothesis that bigger brains correlate with increased learning capabilities. This hypothesis has previously been supported in mammals and birds, and the experiments outlined here provide support for this hypothesis in bees. Overall, the authors present a very clear, well-thought out experiment that adds significantly to the field. In particular, I think the methods of this experiment nicely allow for the behavioral comparisons of many species, which has been a challenge in this area of study. The authors also discuss possible reasons for why some species were unable to be tested in this paradigm, allowing for further comparative studies. I have only a few small comments.

First of all, the choice to weigh brain tissue after fixation is an interesting one – why did the authors choose to do so? Fixation causes shrinkage of tissue and as far as I know, most brain weight measurements in the literature use fresh tissue (e.g. Seid, Castillo, and Wcislo 2011, and Sheehan et al. 2019). This being said, if all brains were treated the same way, the relative differences in weights are likely still valid, as long as all tissue was fixed the same amount. However, this assumes that all species' brains shrink the same amount. This is an assumption that is often made when staining tissue for immunohistochemistry for measure brain region volumes, although the authors should note these assumptions in their manuscript.

The authors divide up the bee species into two groups: bigger and smaller brains. What are the criteria for each group? I assume these designations are different for the absolute and relative mass measurements – this information should be mentioned in the methods and possibly incorporated into a table or figure.

Along these same lines, it also would be useful to have a table (even in the supplemental materials) listing species and the data associated with each (e.g. brain/body size and learning capability). This could be incorporated with the information noted above regarding bigger and smaller brain groups.

Additional lines comments:

Page 4, Lines 17-18: citations are needed to support this statement

Page 7, Line 5: The dimensions of the box should be stated here (2.5 x 13 cm)

Page 9, Line 19: This should be Figure 2, as this figure is discussed second in the manuscript.

Page 12, Line 24: It seems as though there is a line of text missing here.

Page 13, Line 7: A citation for the statement that “larger bees have larger visual organs and higher mobility” is needed.

Page 13, Lines 8-10: This sentence seems to contradict the authors' findings that the bees with larger relative brain size did not decrease their latency to choosing as quickly as those with smaller relative brain size.

Page 14, Lines 16-17: Which of the bees studied here are considered social? This information would be nice to have in this discussion.

Review form: Reviewer 2

Is the manuscript scientifically sound in its present form?

Yes

Are the interpretations and conclusions justified by the results?

Yes

Is the language acceptable?

Yes

Do you have any ethical concerns with this paper?

No

Have you any concerns about statistical analyses in this paper?

No

Recommendation?

Accept with minor revision (please list in comments)

Comments to the Author(s)

General comments:

In this study Collado and colleagues investigate whether insects with bigger brains perform better in an associative learning task. They show that absolute and relative brain size relate to better performance in the task. Their data complements similar findings in birds and mammals. The experiments are done rigorously.

Enthusiasm is somewhat mitigated by the fact that except for the brain size analysis there is no further discussion of the studied species in relation to ecological niche and brain size. Two of the four analyzed families seem not to learn the task which should be further discussed. The learning task which they use is designed for honey bees. This makes honey bees by default good at performing. The families with the smaller sized brains however do not learn the task/perform worse. This might not be related to brain size but rather to their ecological niche that makes this kind of color learning less important. I wonder within honey bees and bumble bees, is the correlation bigger brains learn better still true? These points should be discussed.

However, this study provides sufficient data that add to the existing body of work in vertebrates, which makes the project of interest to the scientific community.

Specific points:

Please discuss the huge variation in brain size within families. Is the variation similar within species? A figure would help here.

Please consider revising your manuscript with respect to species/families. It seems like you are using the term species but give names of insect families (Apidae).

Please discuss that 2 of the 4 species did not seem to learn the task (Andrenidae, Halictidae), figure 4a). See general comment.

Please add number of animals per species to figure legends.

P4 L21: Maybe add than insects at the end? Sentence reads incomplete.

P4 L22/L23: A bit exaggerated.

P5 L1: s missing (obtains)

P5 L16: reads odd since you are making the case that a bigger brain might not have an advantage over a small brain and now you say the benefits of a small brain. And the whole paper tries to prove that bigger brains are better.

P6 L12: Citation: Laloir vs Laloï on Page 23

P6 L12: This is only true for bumble bees and honey bees. What about the solitary bees in your study?

P7 L2: I would recommend moving the relevant parts of table S2 into the main text and adding numbers of animals into figure legends. Reading the manuscript, it would be nice to know how many animals you have per species.

P7 L13: How certain are you the different bee species have the same receptors and can distinguish the colors? This might also be one reason some of the solitary bees fail. See Briscoe and Chittka 2001.

P7: Did you change the rewarded color between animals?

P8 L22: Please discuss change in size due to fixation with PFA (see also Wehrl et al., 2014). Since you are weighing the brains it might not matter as much but still, different shrinkage might impact measurements.

P10 L3 controlling for not by

P11 Please make sure to add all figure numbers where you discuss data also plotted in the figures.

P11 L12/13 It is unclear whether you discuss the training or test trials here.

P11 L22/23 among vs across species. Please consider using a different wording here, it is not clear what you mean, I assume across species?

P12 L8: 'but we found the opposite effect' please elaborate. This is only clear if you also study the figure.

P12: Please consider adding figure numbers to the discussion as well.

P12 L24: end of sentence missing?

P12 L15: 'This supports the view that they learn faster' This sounds misleading.

P14 L15: I am in general missing a discussion of the ecological background of the species in connection to their brain size as well as a mentioning of the discussed species by name.

P16 L12: consider rewriting sentence, it is hard to understand.

P17 Figure 5. Please add explanation for Boxplot

P28: Images are too dark. It is hard to see the animal.

P28 L46: Holes were done, use drilled? Extremes-use sides?

P31: Please add in mg to labels.

Decision letter (RSOS-201940.R0)

Dear Dr Collado

The Editors assigned to your paper RSOS-201940 "Brain size predicts learning abilities in bees" have now received comments from reviewers and would like you to revise the paper in accordance with the reviewer comments and any comments from the Editors. Please note this decision does not guarantee eventual acceptance.

We invite you to respond to the comments supplied below and revise your manuscript. Below the referees' and Editors' comments (where applicable) we provide additional requirements.

Final acceptance of your manuscript is dependent on these requirements being met. We provide guidance below to help you prepare your revision.

Please submit your revised manuscript and required files (see below) no later than 21 days from today's (ie 10-Feb-2021) date. Note: the ScholarOne system will 'lock' if submission of the revision is attempted 21 or more days after the deadline. If you do not think you will be able to meet this deadline please contact the editorial office immediately.

on behalf of Dr Ryan Y Wong (Associate Editor) and Kevin Padian (Subject Editor)
openscience@royalsociety.org

Associate Editor Comments to Author (Dr Ryan Y Wong):
Comments to the Author:

Your manuscript has now been reviewed by two reviewers. Both reviewers agree that the experiment is interesting and provides much needed data in the field. However, several concerns were raised that need to be addressed prior to further consideration. In particular, there were concerns about clarity of methodological procedures and applicability of data interpretation given the experimental design.

Reviewer comments to Author:
Reviewer: 1

Comments to the Author(s)

This manuscript addresses the longstanding hypothesis that bigger brains correlate with increased learning capabilities. This hypothesis has previously been supported in mammals and birds, and the experiments outlined here provide support for this hypothesis in bees. Overall, the authors present a very clear, well-thought out experiment that adds significantly to the field. In particular, I think the methods of this experiment nicely allow for the behavioral comparisons of many species, which has been a challenge in this area of study. The authors also discuss possible reasons for why some species were unable to be tested in this paradigm, allowing for further comparative studies. I have only a few small comments.

First of all, the choice to weigh brain tissue after fixation is an interesting one – why did the authors choose to do so? Fixation causes shrinkage of tissue and as far as I know, most brain weight measurements in the literature use fresh tissue (e.g. Seid, Castillo, and Wcislo 2011, and Sheehan et al. 2019). This being said, if all brains were treated the same way, the relative differences in weights are likely still valid, as long as all tissue was fixed the same amount.

However, this assumes that all species' brains shrink the same amount. This is an assumption that is often made when staining tissue for immunohistochemistry for measure brain region volumes, although the authors should note these assumptions in their manuscript.

The authors divide up the bee species into two groups: bigger and smaller brains. What are the criteria for each group? I assume these designations are different for the absolute and relative mass measurements – this information should be mentioned in the methods and possibly incorporated into a table or figure.

Along these same lines, it also would be useful to have a table (even in the supplemental materials) listing species and the data associated with each (e.g. brain/body size and learning capability). This could be incorporated with the information noted above regarding bigger and smaller brain groups.

Additional lines comments:

Page 4, Lines 17-18: citations are needed to support this statement

Page 7, Line 5: The dimensions of the box should be stated here (2.5 x 13 cm)

Page 9, Line 19: This should be Figure 2, as this figure is discussed second in the manuscript.

Page 12, Line 24: It seems as though there is a line of text missing here.

Page 13, Line 7: A citation for the statement that “larger bees have larger visual organs and higher mobility” is needed.

Page 13, Lines 8-10: This sentence seems to contradict the authors' findings that the bees with larger relative brain size did not decrease their latency to choosing as quickly as those with smaller relative brain size.

Page 14, Lines 16-17: Which of the bees studied here are considered social? This information would be nice to have in this discussion.

Reviewer: 2

Comments to the Author(s)

General comments:

In this study Collado and colleagues investigate whether insects with bigger brains perform better in an associative learning task. They show that absolute and relative brain size relate to better performance in the task. Their data complements similar findings in birds and mammals. The experiments are done rigorously.

Enthusiasm is somewhat mitigated by the fact that except for the brain size analysis there is no further discussion of the studied species in relation to ecological niche and brain size. Two of the four analyzed families seem not to learn the task which should be further discussed. The learning task which they use is designed for honey bees. This makes honey bees by default good at

performing. The families with the smaller sized brains however do not learn the task/perform worse. This might not be related to brain size but rather to their ecological niche that makes this kind of color learning less important. I wonder within honey bees and bumble bees, is the correlation bigger brains learn better still true? These points should be discussed.

However, this study provides sufficient data that add to the existing body of work in vertebrates, which makes the project of interest to the scientific community.

Specific points:

Please discuss the huge variation in brain size within families. Is the variation similar within species? A figure would help here.

Please consider revising your manuscript with respect to species/families. It seems like you are using the term species but give names of insect families (Apidae).

Please discuss that 2 of the 4 species did not seem to learn the task (Andrenidae, Halictidae), figure 4a). See general comment.

Please add number of animals per species to figure legends.

P4 L21: Maybe add than insects at the end? Sentence reads incomplete.

P4 L22/L23: A bit exaggerated.

P5 L1: s missing (obtains)

P5 L16: reads odd since you are making the case that a bigger brain might not have an advantage over a small brain and now you say the benefits of a small brain. And the whole paper tries to prove that bigger brains are better.

P6 L12: Citation: Laloir vs Laloï on Page 23

P6 L12: This is only true for bumble bees and honey bees. What about the solitary bees in your study?

P7 L2: I would recommend moving the relevant parts of table S2 into the main text and adding numbers of animals into figure legends. Reading the manuscript, it would be nice to know how many animals you have per species.

P7 L13: How certain are you the different bee species have the same receptors and can distinguish the colors? This might also be one reason some of the solitary bees fail. See Briscoe and Chittka 2001.

P7: Did you change the rewarded color between animals?

P8 L22: Please discuss change in size due to fixation with PFA (see also Wehrl et al., 2014). Since you are weighing the brains it might not matter as much but still, different shrinkage might impact measurements.

P10 L3 controlling for not by

P11 Please make sure to add all figure numbers where you discuss data also plotted in the figures.

P11 L12/13 It is unclear whether you discuss the training or test trials here.

P11 L22/23 among vs across species. Please consider using a different wording here, it is not clear what you mean, I assume across species?

P12 L8: 'but we found the opposite effect' please elaborate. This is only clear if you also study the figure.

P12: Please consider adding figure numbers to the discussion as well.

P12 L24: end of sentence missing?

P12 L15: 'This supports the view that they learn faster' This sounds misleading.

P14 L15: I am in general missing a discussion of the ecological background of the species in connection to their brain size as well as a mentioning of the discussed species by name.

P16 L12: consider rewriting sentence, it is hard to understand.

P17 Figure 5. Please add explanation for Boxplot

P28: Images are too dark. It is hard to see the animal.

P28 L46: Holes were done, use drilled? Extremes-use sides?

P31: Please add in mg to labels.

===PREPARING YOUR MANUSCRIPT===

===PREPARING YOUR REVISION IN SCHOLARONE===

<https://royalsociety.org/journals/authors/author-guidelines/#supplementary-material> to include a suitable title and informative caption. An example of appropriate titling and captioning may be found at https://figshare.com/articles/Table_S2_from_Is_there_a_trade-off_between_peak_performance_and_performance_breadth_across_temperatures_for_aerobic_sc_ope_in_teleost_fishes_/3843624.

Author's Response to Decision Letter for (RSOS-201940.R0)

See Appendix A.

RSOS-201940.R1 (Revision)

Review form: Reviewer 1

Is the manuscript scientifically sound in its present form?

Yes

Are the interpretations and conclusions justified by the results?

Yes

Is the language acceptable?

Yes

Do you have any ethical concerns with this paper?

Yes

Have you any concerns about statistical analyses in this paper?

No

Recommendation?

Accept with minor revision (please list in comments)

Comments to the Author(s)

The authors have thoroughly revised their manuscript and have addressed all of my comments. In particular, the addition of Table S2 is helpful for understanding the behavioral results more fully. However, a few minor issues remain.

As I mentioned previously, in Figure 5, the authors divide up the bee species into two groups: bigger and smaller brains. The authors clarified that brain size was used as a continuous variable in these analyses. If this is the case, however, I'm confused as to why there are two lines in this graph, one for bigger and one for smaller brains. This needs to be further clarified because the figure does not seem to match the text.

Page 11, Line 12: "Although" instead of "despite"

Page 13, Line 9: How long after the training trials was the test done?

Page 18, Line 17: I think the phrase "solving the learning" should be replaced by performing the learning task successfully or something similar.

Page 19, Line 21: Refer to supplemental table S2 here

Figure S1 is not visible (I'm not sure if this is a problem solely on my end).

Review form: Reviewer 2

Is the manuscript scientifically sound in its present form?

Yes

Are the interpretations and conclusions justified by the results?

Yes

Is the language acceptable?

Yes

Do you have any ethical concerns with this paper?

No

Have you any concerns about statistical analyses in this paper?

No

Recommendation?

Accept as is

Comments to the Author(s)

All concerns and comments have been addressed by the authors.

Decision letter (RSOS-201940.R1)

Dear Dr Collado

On behalf of the Editors, we are pleased to inform you that your Manuscript RSOS-201940.R1 "Brain size predicts learning abilities in bees" has been accepted for publication in Royal Society Open Science subject to minor revision in accordance with the referees' reports. Please find the referees' comments along with any feedback from the Editors below my signature.

Please submit your revised manuscript and required files (see below) no later than 7 days from today's (ie 23-Mar-2021) date. Note: the ScholarOne system will 'lock' if submission of the revision is attempted 7 or more days after the deadline. If you do not think you will be able to meet this deadline please contact the editorial office immediately.

At this stage, we ask that you please archive your GitHub code within the Zenodo repository: <https://guides.github.com/activities/citable-code/>. By doing this, a formal, citable DOI will be associated with your data record, and an open license (CC-BY preferred) can be applied to your data. We would then ask that you please update your data availability statement to read as:

"Data and relevant code for this research work are stored in GitHub: [GitHub URL here] and have been archived within the Zenodo repository: <https://doi.org/zenodo.....> [ref number].

Kind regards,

Royal Society Open Science Editorial Office
Royal Society Open Science

on behalf of Dr Ryan Y Wong (Associate Editor) and Kevin Padian (Subject Editor)
openscience@royalsociety.org

Associate Editor Comments to Author (Dr Ryan Y Wong):

Associate Editor: 1

Comments to the Author:

Thank you for addressing the concerns of the reviewers. Several minor concerns remain that should be addressed.

Reviewer comments to Author:

Reviewer: 1

Comments to the Author(s)

The authors have thoroughly revised their manuscript and have addressed all of my comments. In particular, the addition of Table S2 is helpful for understanding the behavioral results more fully. However, a few minor issues remain.

As I mentioned previously, in Figure 5, the authors divide up the bee species into two groups: bigger and smaller brains. The authors clarified that brain size was used as a continuous variable in these analyses. If this is the case, however, I'm confused as to why there are two lines in this graph, one for bigger and one for smaller brains. This needs to be further clarified because the figure does not seem to match the text.

Page 11, Line 12: "Although" instead of "despite"

Page 13, Line 9: How long after the training trials was the test done?

Page 18, Line 17: I think the phrase "solving the learning" should be replaced by performing the learning task successfully or something similar.

Page 19, Line 21: Refer to supplemental table S2 here

Figure S1 is not visible (I'm not sure if this is a problem solely on my end).

Reviewer: 2

Comments to the Author(s)

All concerns and comments have been addressed by the authors.

===PREPARING YOUR MANUSCRIPT===

Your revised paper should include the changes requested by the referees and Editors of your manuscript. You should provide two versions of this manuscript and both versions must be provided in an editable format:
one version identifying all the changes that have been made (for instance, in coloured highlight, in bold text, or tracked changes);
a 'clean' version of the new manuscript that incorporates the changes made, but does not highlight them. This version will be used for typesetting.

===PREPARING YOUR REVISION IN SCHOLARONE===

- If you are providing image files for potential cover images, please upload these at this step, and inform the editorial office you have done so. You must hold the copyright to any image provided.
- A copy of your point-by-point response to referees and Editors. This will expedite the preparation of your proof.

- Ensure that your data access statement meets the requirements at <https://royalsociety.org/journals/authors/author-guidelines/#data>. You should ensure that you cite the dataset in your reference list. If you have deposited data etc in the Dryad repository, please only include the 'For publication' link at this stage. You should remove the 'For review' link.
- If you are requesting an article processing charge waiver, you must select the relevant waiver option (if requesting a discretionary waiver, the form should have been uploaded at Step 3 'File upload' above).
- If you have uploaded ESM files, please ensure you follow the guidance at <https://royalsociety.org/journals/authors/author-guidelines/#supplementary-material> to include a suitable title and informative caption. An example of appropriate titling and captioning may be found at https://figshare.com/articles/Table_S2_from_Is_there_a_trade-off_between_peak_performance_and_performance_breadth_across_temperatures_for_aerobic_scope_in_teleost_fishes_/3843624.

Author's Response to Decision Letter for (RSOS-201940.R1)

See Appendix B.

Decision letter (RSOS-201940.R2)

Dear Dr Collado,

It is a pleasure to accept your manuscript entitled "Brain size predicts learning abilities in bees" in its current form for publication in Royal Society Open Science. The comments of the reviewer(s) who reviewed your manuscript are included at the foot of this letter.

on behalf of Dr Ryan Y Wong (Associate Editor) and Kevin Padian (Subject Editor)
openscience@royalsociety.org

Subject Editor Comments to Author (Professor Kevin Padian):

Comments to the Author:

Thanks for your attention to comments in your revision. We're pleased to accept it for publication. Best wishes.

Appendix A

Dear Dr Ryan Y Wong,

Thank you for the swift handling of our manuscript. We have incorporated your insightful comments and added modifications to our article when relevant. We hope that this new manuscript suits the quality standards to be published in Royal Society Open Science. Find below a detailed response point by point to the reviewers comments. We also marked in blue font the main changes done to the main text.

Associate Editor Comments to Author (Dr Ryan Y Wong):

Your manuscript has now been reviewed by two reviewers. Both reviewers agree that the experiment is interesting and provides much needed data in the field. However, several concerns were raised that need to be addressed prior to further consideration. In particular, there were concerns about clarity of methodological procedures and applicability of data interpretation given the experimental design.

Answer: Thank you for your positive assessment. In this new version we clarified the methodological protocol used and clarified what we can interpret and what not.

Reviewer 1 comments to Author:

This manuscript addresses the longstanding hypothesis that bigger brains correlate with increased learning capabilities. This hypothesis has previously been supported in mammals and birds, and the experiments outlined here provide support for this hypothesis in bees. Overall, the authors present a very clear, well-thought out experiment that adds significantly to the field. In particular, I think the methods of this experiment nicely allow for the behavioral comparisons of many species, which has been a challenge in this area of study. The authors also discuss possible reasons for why some species were unable to be tested in this paradigm, allowing for further comparative studies. I have only a few small comments.

Answer: Thank you for your kind words!

First of all, the choice to weigh brain tissue after fixation is an interesting one – why did the authors choose to do so? Fixation causes shrinkage of tissue and as far as I know, most brain weight measurements in the literature use fresh tissue (e.g. Seid, Castillo, and Wcislo 2011, and Sheehan et al. 2019). This being said, if all brains were treated the same way, the relative differences in weights are likely still valid, as long as all tissue was fixed the same amount. However, this assumes that all species' brains shrink the same amount. This is an assumption that is often made when staining tissue for immunohistochemistry for measure brain region volumes, although the authors should note these assumptions in their manuscript.

Answer: Thank you for pointing this out. We added a statement about brains shrinkage (Line 156-157) and clarify the assumptions taken.

Reviewer 1: The authors divide up the bee species into two groups: bigger and smaller brains. What are the criteria for each group? I assume these designations are different for the absolute and relative mass measurements – this information should be mentioned in the methods and possibly incorporated into a table or figure.

Answer: We are sorry if this was not clear enough. All analyses were done with brain size as a continuous variable. As the reviewer suggest, any division between big and small will be arbitrary and hard to justify. We clarified this in text and on the figure captions.

R1: Along these same lines, it also would be useful to have a table (even in the supplemental materials) listing species and the data associated with each (e.g. brain/body size and learning capability). This could be incorporated with the information noted above regarding bigger and smaller brain groups.

Answer: We added a table in the supplementary material with all raw data analyzed.

R1: Page 4, Lines 17-18: citations are needed to support this statement

A: Reference added.

Page 7, Line 5: The dimensions of the box should be stated here (2.5 x 13 cm)

A: We added this information.

Page 9, Line 19: This should be Figure 2, as this figure is discussed second in the manuscript.

A: Changed, thanks for noticing the error.

Page 13, Line 7: A citation for the statement that “larger bees have larger visual organs and higher mobility” is needed.

A: Reference added.

Page 13, Lines 8-10: This sentence seems to contradict the authors’ findings that the bees with larger relative brain size did not decrease their latency to choosing as quickly as those with smaller relative brain size.

A: We re-worded the sentence, as the previous version was unclear.

Page 14, Lines 16-17: Which of the bees studied here are considered social? This information would be nice to have in this discussion.

A: We clarified which of our species are social both in text when pertinent, and in the Supplementary Material.

Reviewer 2

In this study Collado and colleagues investigate whether insects with bigger brains perform better in an associative learning task. They show that absolute and relative brain size relate to better performance in the task. Their data complements similar findings in birds and mammals. The experiments are done rigorously. Enthusiasm is somewhat mitigated by the fact that except for the brain size analysis there is no further discussion of the studied species in relation to ecological niche and brain size. Two of the four analyzed families seem not to learn the task which should be further discussed. The learning task

which they use is designed for honey bees. This makes honey bees by default good at performing. The families with the smaller sized brains however do not learn the task/perform worse. This might not be related to brain size but rather to their ecological niche that makes this kind of color learning less important. I wonder within honey bees and bumble bees, is the correlation bigger brains learn better still true? These points should be discussed.

A: Thank you for the encouragement. We share your concerns regarding what drives this pattern and we agree further research should deep into this questions. To adress the concern that a few taxa may drive the response, we use a phylogenitiacly comparative analysis framework. This allowed us to control for the non independency of the different species. Moreover, note that the analysis with relative brain size, bumblebees are not the ones with larger relative brain sizes, and some small species have a large relative brain size, so we do not expect a spurious correlation between ecological niche and relative brain size.

However, this study provides sufficient data that add to the existing body of work in vertebrates, which makes the project of interest to the scientific community.

A: Thank you!

Specific points:

R: Please discuss the huge variation in brain size within families. Is the variation similar within species? A figure would help here.

A: We show in Sup Mat figure S1 a new figure showing the dispersion per families using boxplots. We show that the dispersion is not high within groups and that families are well diferenced by brain size

R: Please consider revising your manuscript with respect to species/families. It seems like you are using the term species but give names of insect families (Apidae).

A: We are not sure what the reviewer refers to. We use species level taxonomy for all analysis. We revised the text to ensure we are clear.

R: Please discuss that 2 of the 4 species did not seem to learn the task (Andrenidae, Halictidae), figure 4a). See general comment.

A: We added this discussion to lines 283-286. Note, however, that with the families Andrenidae and Halictidae, some species do learn, but not others.

R: Please add number of animals per species to figure legends.

A: Added in the figure 4

R: P4 L21: Maybe add than insects at the end? Sentence reads incomplete.

A: Added

P4 L22/L23: A bit exaggerated.

A: We removed this sentence.

P6 L12: This is only true for bumble bees and honey bees. What about the solitary bees in your study?

A: We modified the sentence to clarify that both solitary and social species benefit from learning.

P7 L13: How certain are you the different bee species have the same receptors and can distinguish the colors? This might also be one reason some of the solitary bees fail. See Briscoe and Chittka 2001.

A: This is a very good question. Color vision has been insufficiently studied in solitary bees. However, blue and yellow are colors used in classic experiments with bees, including different families (Apidea, Megachilidae), and hence we assume that they provide enough contrast to be discriminated by all species even when they show differences in photoreceptors. In the new version, we explain this in line 124.

P7: Did you change the rewarded color between animals?

A: Yes, colors were randomized between animals. We state that in line 130-131

P8 L22: Please discuss change in size due to fixation with PFA (see also Wehrl et al., 2014). Since you are weighing the brains it might not matter as much but still, different shrinkage might impact measurements.

A: Thank you for this suggestion. As R1 points out, shrinking is a problem that requires some reasonable assumptions, we added a sentence about this in lines 159-162

P11 Please make sure to add all figure numbers where you discuss data also plotted in the figures.

A: We added the figure number when discussed in text.

P11 L12/13 It is unclear whether you discuss the training or test trials here.

A: Clarified, marked in text.

P11 L22/23 among vs across species. Please consider using a different wording here, it is not clear what you mean, I assume across species?

A: Corrected.

P12 L8: 'but we found the opposite effect' please elaborate. This is only clear if you also study the figure.

A: We reworded the sentence for clarity.

P12 L24: end of sentence missing?

A: Yes, end of sentence was missing, we added it again.

P13 L15: 'This supports the view that they learn faster' This sounds misleading.

A: We re-worded this sentence.

P16 L12: consider rewriting sentence, it is hard to understand.

A: we re-wrote the sentence.

P17 Figure 5. Please add explanation for Boxplot

A: We added this description.

P28 L46: Holes were done, use drilled? Extremes-use sides?

A: Thanks, we use no "drilled".

P31: Please add in mg to labels.

A: Done.

Appendix B

Dear editors

Thank you for the swift handling of our manuscript. We have incorporated your insightful comments and added modifications to our article when relevant. We hope that this new manuscript suits the quality standards to be published in Royal Society Open Science. Find below a detailed response point by point to the reviewers comments. We also marked in blue font the main changes done to the main text.

R: To help improve findability and reproducibility, and to ensure robust dissemination and appropriate credit to authors, we ask that all datasets which accompany manuscripts are cited and acknowledged as bibliographic references, and listed in the References list.

We therefore ask that you please ensure that your Dryad dataset record is listed within your main References list; with the final DOI given (in line with our data sharing policies: <https://royalsociety.org/journals/ethics-policies/data-sharing-mining/>). Your data availability statement within the ScholarOne form (and in your manuscript if this is also provided here) should then appear as follows:

"XXXX data are available within the Dryad Digital Repository: <https://doi.org/....> [REF NUMBER]"

Please then add the full citation to the end of your main References list at the end of your manuscript. Data references in Royal Society journals are in the Vancouver style, for example:

A: We added the dryad URL to the data accesibility statemente, and at the end of the references list